# Development and Internal Validation of a Prediction Model for Surgical Success of Maxillomandibular Advancement for the Treatment of Moderate to Severe Obstructive Sleep Apnea

**DOI:** 10.3390/jcm12020503

**Published:** 2023-01-07

**Authors:** Wouter P. Visscher, Jean-Pierre T. F. Ho, Ning Zhou, Madeline J. L. Ravesloot, Engelbert A. J. M. Schulten, Jan de Lange, Naichuan Su

**Affiliations:** 1Amsterdam UMC and Academic Centre for Dentistry Amsterdam (ACTA), Department of Oral and Maxillofacial Surgery/Oral Pathology, University of Amsterdam, 1105 AZ Amsterdam, The Netherlands; 2Department of Oral and Maxillofacial Surgery, Noordwest Ziekenhuisgroep, 1815 JD Alkmaar, The Netherlands; 3Academic Centre for Dentistry Amsterdam (ACTA), Department of Orofacial Pain and Dysfunction, University of Amsterdam and Vrije Universiteit Amsterdam, 1081 LA Amsterdam, The Netherlands; 4Department Otorhinolaryngology—Head and Neck Surgery, OLVG, 1061 AE Amsterdam, The Netherlands; 5Amsterdam UMC and Academic Centre for Dentistry Amsterdam (ACTA), Department of Oral and Maxillofacial Surgery/Oral Pathology, Vrije Universiteit Amsterdam, 1081 HV Amsterdam, The Netherlands; 6Academic Centre for Dentistry Amsterdam (ACTA), Department of Oral Public Health, University of Amsterdam and Vrije Universiteit Amsterdam, 1081 LA Amsterdam, The Netherlands

**Keywords:** obstructive sleep apnea, maxillomandibular advancement, prediction, surgical success

## Abstract

Background: Maxillomandibular advancement (MMA) has been shown to be the most effective surgical therapy for obstructive sleep apnea (OSA). Despite high success rates, there are patients who are considered as non-responders to MMA. In order to triage and inform these patients on their expected prognosis of MMA before the surgery, this study aimed to develop, internally validate, and calibrate a prediction model for the presence of surgical success for MMA in patients with OSA. Methods: A retrospective cohort study was conducted that included patients that had undergone MMA for moderate to severe OSA. Baseline clinical, polysomnographic, cephalometric, and drug-induced sleep endoscopy findings were recorded as potential predictors. Presence or absence of surgical success was recorded as outcome. Binary logistic regression analyses were conducted to develop the model. Performance and clinical values of the model were analyzed. Results: One hundred patients were included, of which sixty-seven (67%) patients reached surgical success. Anterior lower face height (ALFH) (OR: 0.93 [0.87–1.00], *p* = 0.05), superior posterior airway space (SPAS) (OR: 0.76 [0.62–0.92], *p* < 0.05), age (OR: 0.96 [0.91–1.01], *p* = 0.13), and a central apnea index (CAI) <5 events/hour sleep (OR: 0.16 [0.03–0.91], *p* < 0.05) were significant independent predictors in the model (significance level set at *p* = 0.20). The model showed acceptable discrimination with a shrunken area under the curve of 0.74, and acceptable calibration. The added predictive values for ruling in and out of surgical success were 0.21 and 0.32, respectively. Conclusions: Lower age at surgery, CAI < 5 events/hour, lower ALFH, and smaller SPAS were significant predictors for the surgical success of MMA. The discrimination, calibration, and clinical added values of the model were acceptable.

## 1. Introduction

Obstructive sleep apnea (OSA) is a breathing disorder which occurs during sleep and is characterized by recurrent obstruction (partial or complete) of the upper airway, resulting in hypopnea and/or apnea [1]. OSA results in hypoxemia, hypercapnia, and arousals from sleep. It is associated with cardiovascular and cognitive morbidity, a reduced quality of life, and premature death [2,3,4,5,6]. It is estimated that the prevalence of OSA in the general population is 9% to 38%, whilst prevalence percentages increase due to rising rates of obesity in addition to an aging population [7,8]. Polysomnography (PSG) is the gold standard test for the diagnosis of OSA. The diagnosis and severity of OSA have been largely quantified by the numeric calculation of the number of obstructive, central, and mixed apneas and hypopneas per hour of sleep (AHI). Severity, spanning three levels, is traditionally defined by the cut-offs 5–14, 15–29 and ≥30 events per hour defining mild, moderate and severe OSA, respectively, as suggested by the American Society of Sleep Medicine (AASM) [9].

Continuous positive airway pressure (CPAP) is considered the first treatment choice in patients with moderate to severe OSA [9]. However, a substantial proportion of patients experience problems tolerating CPAP, resulting in a reduced compliance to the therapy [10]. Alternatives for these patients usually consist of a mandibular advancement device (MAD) or surgical treatment, e.g., maxillomandibular advancement osteotomy (MMA) [11]. MMA has shown to be the most effective surgical therapy for OSA, excluding a tracheostomy, with a reported success rate of 85% [12]. However, despite the high success rates, there is a group of patients who are considered as non-responders to MMA [12]. It is thought that the presence of complete anteroposterior collapse at the level of the epiglottis and a minimal retro velar space might contribute to MMA failure [13,14]. However, only a few studies have assessed predictors for failure in MMA; therefore, drawing conclusions remains arbitrary.

In order to efficiently use the scarce medical resources, it is of utmost importance to triage the patients based on their expected prognosis of MMA before the surgeries. To ensure this, prediction models for surgical success are of vital importance. To date, no prediction models for the surgical success of MMA have been developed, further complicating preoperative clinical patient counseling and suitable candidate selection. This is because a prediction model helps to inform patients on their potential prognosis of the surgery and also aids clinicians during preoperative decision-making. Therefore, prediction models for the surgical success of MMA are warranted. Whilst we nowadays aim for tailor-made treatment (personalized medicine) for each individual patient, it is important that preoperative predictors for surgical success are identified. These predictors should lead to the development, validation, and implementation of a prediction model for the surgical success of MMA as a treatment of OSA in the future. Improving MMA candidate selection will not only contribute to improve appropriate care delivery, but also reduce morbidity and increase the therapeutic success of MMA. A broader goal is to better utilize the available healthcare costs by optimizing the cost-effectiveness of MMA as a treatment for OSA. Therefore, the aim of this study was to identify potential predictors for the surgical success of MMA (as defined by Sher’s criteria [15]) in patients with OSA, and develop and internally validate a model for the prediction of surgical success.

## 2. Materials and Methods

The Medical Ethics Committee of the Amsterdam University Medical Centers (Amsterdam UMC, location Amsterdam Medical Center (AMC)) concluded that this study was exempted from the Medical Research Human Subjects Act (Reference number W22_061#22.093). The present study was carried out based on the Strengthening The Reporting of Observational studies in Epidemiology (STROBE) [16] statement and the Transparent Reporting of a multivariable prediction model for Individual Prognosis or Diagnosis (TRIPOD) statement [17].

### 2.1. Study Design and Participants Enrolment

The study was designed as a retrospective cohort study. The inclusion criteria were (1) patients with moderate to severe OSA, diagnosed by means of PSG (AHI ≥ 15/h); (2) age > 18 years old; (3) patients who underwent MMA as a treatment for OSA in the Amsterdam UMC location AMC, from September 2011 to September 2020; (4) an overnight level I or level II PSG was performed to measure the parameters relevant to OSA prior to surgery and at a minimum of 3 months postoperatively; (5) a standardized lateral cephalogram was performed prior to surgery and at a minimum of one week postoperatively; and (6) patients who were followed-up for at least 12 months on the outpatient clinic after MMA.

The non-inclusion criteria were as follows: (1) patients who did not undergo isolated MMA nor simultaneous upper airway surgery (e.g., uvulopalatopharyngoplasty, lateral pharyngoplasty, expansion sphincter pharyngoplasty, barbed reposition pharyngoplasty, tongue volume reduction surgery and/or hyoid bone suspension surgery); (2) patients who underwent a previous MMA osteotomy as a treatment for OSA; (3) patients with instable endocrine dysfunction prior to surgery (hypothyroidism, acromegaly and pituitary adenoma) and/or patients with craniofacial syndromes; and (4) patients who did not give permission for their data to be used for research purposes.

### 2.2. Treatment Protocol

All MMA osteotomies were performed by two experienced oral and maxillofacial surgeons dedicated to the treatment of OSA. MMA osteotomy consisted of a Le Fort I osteotomy of the maxilla with a Hunsuck-Dal Pont modification of the bilateral sagital split osteotomy (BSSO) of the mandible, as described by Obwegeser [18,19]. Subsequently, advancement of the maxillomandibular complex followed, and in a subgroup of patients additional counterclockwise rotation was performed [20]. After applying temporary maxillomandibular fixation by steel-wire ligatures or power chains and intraoperative splints, rigid internal fixation was applied [21,22]. Before the availability of three-dimensional planning, the surgery was planned two-dimensionally with manually fabricated intraoperative splints. In patients who had undergone more recent surgery, the surgery was virtually planned and involved three-dimensionally fabricated intraoperative splints [11].

### 2.3. Predictors

The potential predictors were extracted from the electronic patients’ files, including patient-related variables, respiratory parameters assessed by PSG, drug-induced sleep endoscopy (DISE) findings, and cephalometric measurements. All the predictors were measured at baseline before the MMA. All the potential predictors in the present study were decided based on the previous literature [11,23,24] and the authors’ clinical experience and knowledge.

#### 2.3.1. Patient-Related Variables

The patient-related variables included gender, age, body mass index (BMI) at time of surgery, pre-existent physiological status by means of the ASA (American Society of Anesthesiology) classification score (ASA I, normal health; ASA II, mild systemic disease; ASA III, severe systemic disease; ASA IV, severe systemic disease that is a constant threat to life; ASA V, not expected to survive without operation) [25], history of upper airway surgery, excluding previous MMA, as a treatment for OSA (Yes or No), and the presence or absence of teeth (dentulous versus edentulous). Patients with 1–27 teeth (excluding the third molars) were classified as partially dentulous.

#### 2.3.2. Respiratory Parameters

All patients underwent an overnight level I or level II PSG prior to surgery and a minimum of 3 months postoperatively. For scoring respiratory events, we adhered to the criteria of the American Academy of Sleep Medicine (AASM), with the use of the recommend rules for the scoring of hypopneas, i.e., (1) peak signal excursions drop by ≥30% of pre-event baseline using nasal pressure (diagnostic study); (2) the duration of the ≥30% drop in signal excursion is ≥10 s; and (3) ≥3% oxygen desaturation from pre-event baseline and/or the event is associated with an arousal) [26]. The following data was obtained from PSG prior to surgery (baseline): AHI, central apnea index (CAI; presence of central apnea events was defined as a CAI ≥ 5 per hour sleep [27]), and presence of positional OSA (positional OSA was defined as a minimally two times higher AHI in supine position when compared to non-supine position [28]).

#### 2.3.3. Cephalometric Variables

The lateral cephalograms were taken with the patients’ head in a natural position with the mandibular condyle positioned in centric relation to the glenoid fossa. All cephalograms were analyzed by a single observer using Viewbox software (Viewbox 4, dHAL Software, Kifissia, Greece) [29]. For intra-observer reliability analyses, the observer repeated the measurements one month later in twenty cases that were randomly selected. In the present study, the following cephalometric data at baseline was obtained as the potential predictors: anterior lower face height, anterior total face height, presence of maxillomandibular deficiency (maxillomandibular deficiency was defined as sella-naison-A-point (SNA) angle ≤ 80.5° and/or sella-naison-B-point (SNB) angle ≤ 78.5°) [30], and superior posterior airway space (SPAS). An overview on the cephalometric variables and definitions is illustrated in Table 1. An overview of the landmarks, reference lines, and variables on cephalometry is illustrated in Figure 1.

#### 2.3.4. Drug-induced Sleep Endoscopy

In patients with previous unsuccessful CPAP and/or MAD therapy, DISE was performed prior to MMA osteotomy to assess the precise anatomic level(s) and pattern(s) of upper airway collapse. These patients underwent a standardized DISE procedure, of which the method is described in a previous study [27]. In order to quantify the observers’ findings during DISE, the VOTE scoring system was used [28]. In the present study, we included data on presence/absence of concentric collapse at the velum and presence/absence of complete anteroposterior epiglottis collapse, both in supine position, as the potential predictors.

### 2.4. Outcomes

Changes in AHI at 3 to 12 months follow-up compared with the preoperative AHI were regarded as the primary outcome for surgical success. The outcome for surgical success was binary. The surgical success of MMA is considered ‘present’ if a patient’s AHI was reduced by ≥50% compared to the preoperative AHI, combined with a postoperative AHI < 20 events/h, as proposed by Sher et al. [15].

### 2.5. Statistical Analysis

#### 2.5.1. Missing Data

The multiple imputation technique was used for the missing values. We created m = 35 imputed datasets with 10 iterations and used predictive mean matching (PMM) for imputing the missing values. All the potential predictors and the outcome variable were included in the imputation model.

#### 2.5.2. Development of the Model

##### Screening of Potential Predictors and Modelling

The potential predictors for surgical success were determined based on clinical experience and previous literature by the research team. Multicollinearity of the potential predictors were assessed using the variance inflation factor (VIF). When a VIF value of a predictor was higher than 10 [31], collinearity was considered present and the predictor was excluded from the subsequent analysis.

To pre-screen the potential predictors, univariate binary logistic regression analysis was used to assess the association between each potential predictor and the outcome. The predictors with a *p*-value of ≤0.20 were selected for the subsequent multivariate analyses. Multivariate binary logistic regression analysis with backward selection (predictors with *p*-value of >0.20 were removed) was performed to further screen the potential predictors and develop the prediction model.

##### Shrinkage Factor

A global shrinkage factor was produced based on the bootstrapping procedure with 100 bootstrap samples. The shrinkage factor was used to shrink the regression coefficients of the predictors in order to prevent the overfitting of the prediction model [32,33].

##### Performance of the Prediction Model

The performance of the prediction model was assessed in aspects of calibration and discrimination. Calibration is defined as the agreement between predicted and observed outcomes [34]. The calibration of the model was assessed with the calibration plot by plotting the predicted individual outcomes against the observed actual outcomes. The patients were grouped into deciles based on their predicted probabilities of the outcomes. The prevalence of the outcome events in each decile is considered the observed probability. The mean of the individual predicted probabilities in each decile is considered the predicted probability. In the calibration plot, the agreement between predicted probabilities and observed probabilities across the range of the predicted risks was estimated. The overall calibration of the model was assessed with the overall observed–expected ratio (O:E ratio) [34]. The O:E ratio was defined as the ratio between the prevalence of the outcomes (observed) and the mean individual predicted probabilities of the outcomes (expected) within the cohort [35]. An O:E ratio between 0.8 and 1.2 indicates an acceptable overall calibration [36]. The calibration of the model was also assessed with the Hosmer–Lemeshow goodness-of-fit statistic test (HL test). A *p*-value of >0.10 of the HL test indicates that the model fits the observed data [37].

Discrimination is defined as the ability of the model to differentiate between those with and without the outcome events [34]. The discrimination of the model was assessed with the area under the receiver-operating characteristic curve (AUC). An AUC of 0.70 to 0.80 indicates an acceptable discrimination of the model, while an AUC of ≥0.80 indicates an excellent to outstanding discrimination of the model [38].

The optimal cutoff for the predicted probability of the model was defined as the predicted probability with the maximum sum of sensitivity and specificity in the receiver-operating characteristic curve (ROC).

##### Clinical (Added) Values

The clinical values of the model at the optimal cutoff for predicted probability were assessed using prevalence (prior probability) and posterior probabilities of the outcome events. The posterior probability was defined as positive predictive value (PPV) and negative predictive value (NPV). PPV was defined as the number of patients with the actual outcome events among the patients who were predicted to have the outcome events. NPV was defined as the number of patients without actual outcome events among the patients who were predicted to have no outcome events. The added predictive value of the model for ruling in an increased probability of the outcome events was defined as the PPV minus prevalence, while that for ruling out an increased probability of the outcome events was defined as the NPV minus complement of prevalence.

##### Score Chart and Line Chart

A clinical prediction rule for the outcome events was developed to provide an estimate for individual patients of their absolute probability of the outcome events. For the final multivariate binary logistic regression model, the individual probability (P) of the outcome events was predicted with the following formula:P = 1 − 1/[1 + exp(constant + β1X1 + … + βiXi)]
where β is the shrunken regression coefficient of a predictor in the models.

To facilitate the calculation of the predicted probability of the outcome events in individual patients, the multivariate logistic regression model was converted to a score chart. In the score chart, the score of each included predictor was produced by the shrunken regression coefficients being multiplied by −100 and subsequently rounded. A line chart was then developed to help determine the predicted probability of the outcome events.

All the statistical procedures mentioned above were performed via SPSS 27.0 (IBM, New York, NY, USA) and R software 4.0.4 ((R Development Core Team, Vienna, Austria).

## 3. Results

In the period of September 2011 to September 2020, 111 patients underwent MMA osteotomy for OSA. A total of 100 patients were eligible for analysis, of whom 82 (82%) were male. Eleven patients were excluded due to no patient approval for usage of their data for research purposes (*n* = 3), mild OSA (*n* = 3), no postoperative PSG performed (*n* = 4), and craniofacial syndrome (*n* = 1). Among the 100 eligible patients, mean age was 50.5 (± 9.9) years and mean BMI was 29.8 (±4.2) kg/m^2^. The majority of patients were ASA II (56%), followed by ASA I (23%) and ASA III (21%). In ninety-eight (98%) patients, CPAP was an unsuccessful therapy and/or intolerance was noted. Two (2%) patients declined CPAP as first-choice therapy. Mean AHI prior to surgery was 52.9 (± 21.4), and 16 (16%) patients had a CAI of ≥ 5 events per hour of sleep. A total of 67 (67%) patients had surgical success from treatment. The median preoperative percentage of total sleep time spent in supine position in the total population, the surgical success subgroup, and the surgical failure subgroup was 37.3% (interquartile range [IQR], 19.0–56.0), 36.3% (IQR, 16.7–56.1), and 44.0% (IQR, 25.5–54.6), respectively; after MMA, they were 37.0% (IQR, 17.0–53.0), 30.0% (IQR, 10.3–49.4), and 40.5% (IQR, 28.6–58.2), respectively. The median preoperative percentage of total sleep time spent in the rapid eye movement (REM) stage was 17.8% (IQR, 12.1–21.5), 17.8% (IQR, 12.1–21.5), and 18.3% (IQR, 12.1–21.7), respectively; after MMA, they were 22.5% (IQR, 15.8–27.2), 24.0% (IQR, 17.4–29.1), and 19.0% (IQR, 13.4–25.8), respectively. Preoperatively, the median ODI 3% in the total population, surgical success subgroup, and surgical failure subgroup was 51.0 (IQR, 34.4–66.6) events/hour, 48.7 (IQR, 35.3–68.9) events/hour, and 57.0 (IQR, 29.5–66.0) events/hour, respectively; postoperatively, they were 21.1 (IQR, 10.5–30.2) events/hour, 11.2 (IQR, 9.2–20.7) events/hour, and 33.6 (IQR, 25.8–50.3) events/hour, respectively. Further details on the baseline characteristics of the potential predictors and their distribution over the outcome are presented in Table 2 (Appendix A contains Table A1, which presents baseline characteristics without multiple imputation).

The VIF values of all the predictors were lower than 10, which indicated that the multicollinearity between the predictors was negligible. Therefore, all the predictors were included for further analysis. In the univariate binary logistic regression analyses, anterior total face height, anterior lower face height, SPAS, age, and presence of CAI ≥ 5 events/hour had a *p*-values of ≤0.20 and were included in the subsequent multivariate binary logistic regression analysis (Table 3). In the multivariate analysis, anterior lower face height, SPAS, age, and presence of CAI ≥ 5 events/hour remained in the final model with *p*-values of ≤0.20 (Table 3).

The shrinkage factor of the model was 0.80. The original AUC of the model was 0.78 (95% confidence interval [95%CI]: 0.66 to 0.87) and the shrunken AUC of the model was 0.74. This indicated that the discrimination of the model was acceptable. The calibration plot (Figure 2) showed that most plotted dots were lying close to the diagonal line. Therefore, there was a good agreement between the predicted probabilities and actual probabilities of the outcomes. The O:E ratio was 1.01 (95%CI: 0.81 to 1.24), which indicated that the overall calibration of the model was excellent. The *p*-value of the HL test was 0.42, which showed that the model had good fit.

The optimal cutoff for the predicted probability of the model was 0.62. Table 4 presents the prevalence, sensitivity, specificity, PPV, and NPV of the model. The clinical added value of the model for ruling in the probability of surgical success was 0.21 (95%CI: 0.09 to 0.34) in addition to the prevalence, while that for ruling out the probability of surgical success was 0.32 (95%CI: 0.15 to 0.49) in addition to the complement of the prevalence.

To enhance the clinical usefulness of the model, a score chart (Table 5) and a line chart (Figure 3) were produced. A clinician can easily calculate the sum score of a patient using the score chart and determine the corresponding predicted probability of surgical success based on a line chart using the sum score. The predicted probability of surgical success is lower when the sum score is higher. The cutoff of the sum score for the prediction of surgical success was 1111.

The algorithm for the calculation of a patient’s sum score for surgical success is presented below:*Sum score* = 6 ∗ *anterior lower face height* + 23 ∗ *SPAS* + 3 ∗ *age* + 147 ∗ *CAI* ≥ 5 *events/hour*

## 4. Discussion

In the present study, patients with a lower age at surgery, CAI < 5 events per hour, a lower anterior lower face height (ALFH), and a smaller superior posterior airway space (SPAS) may have a higher probability of obtaining surgical success. The prediction model for the surgical success of MMA was derived based on the predictors above, and the performance of the model may be acceptable. To the authors’ best knowledge, this is the first study to develop a prediction model for the surgical success of MMA for the treatment of OSA with pre-operative patient data that can be utilized during daily clinical practice.

Clinicians frequently encounter the presence of central and/or mixed events on PSG in patients with OSA, which makes the treatment decision-making process more difficult [39]. The results presented in this study on the CAI and its role with respect to the surgical success of MMA are in line with a study by Markovey et al. [13], illustrating that a lower pre-operative CAI was a statistically significant predictor of surgical success (CAI pre-operatively in the success group was 0.6 versus 5.7 in the failure group, *p*-value = 0.005). Xie et al. studied the difference between patients with pure OSA (100% of the apneas are obstructive) and predominant OSA (presence of both central and obstructive apneas and the obstructive apneas account for >50% of the total number of apneas), and they reported lower breathing control stability in patients with predominant OSA [40]. Therefore, it is thought that in patients with a higher preoperative CAI, the lower breathing control stability might entail obstructive events, leading to lower surgical success rates. This present study also found that ALFH was significantly associated with surgical success. In a meta-analysis on craniofacial morphology in patients with OSA, the authors found a strong tendency towards an increased ALFH in adult patients with OSA [41]. A possible explanation for this altered craniofacial anatomy might be upper airway obstruction occurring as early as childhood [42]. However, to date, still little is known regarding the exact underlying mechanism of cephalometric measurements as predictors for surgical success. Despite the fact that the included predictors in the prediction model were significantly associated with surgical success, the causality between predictor and outcome was not assessed, and conclusions on causality cannot be drawn. Therefore, included predictors might not have a causal relation, whilst still being strong predictors for surgical success in the prediction model.

The original AUC of the model was 0.78, and the shrunken AUC of the model was 0.74, which indicates that the discrimination of the model was acceptable. The calibration plot (Figure 2) illustrates that there was a good agreement between the predicted probabilities and the actual probabilities of the outcomes. The added predictive value for ruling in surgical success was 0.21, whereas the added predictive value for ruling out surgical success was 0.32. These results denote that if the model predicts a patient to reach surgical success, the posterior probability of such patient to reach surgical success can be increased by 0.21 when compared with the prevalence of surgical success in the patient’s group. If the model predicts a patient to have the absence of surgical success, such patient’s posterior probability of an absence of surgical success can be increased by 0.32 when compared with the completement of prevalence of surgical success in the patient’s group. Both these results denote that the clinical added values of the model were adequate for ruling in and ruling out surgical success.

In order to optimize the utilization of the model during daily clinical practice, calculation of the optimal cut-off value for predicted probability is needed for probability stratification. The optimal cut-off value is determined when both sensitivity and specificity are at their maximum, so false negative and false positive outcomes are at their lowest. The optimal cutoff for the predicted probability of surgical success was 0.62. Thus, in the event of a sum score lower than 1111, individuals were very likely to reach surgical success.

Of note is the fact that a prediction model might entail false positive and false negative outcomes. In the event of a false negative outcome, a patient and clinician might falsely waive MMA as the therapy of choice, which might worsen the patient’s OSA and prognosis. On the other hand, a false positive outcome might lead to an incorrect indication for surgery, which entails comorbidity and the risks associated with surgery, such as bleeding, infection, and wound healing problems. Both false negative and false positive outcomes might result in an increase in costs and unfavorable health outcomes. The model presented in this study has a 35% and 12% risk of a false negative and false positive outcome, respectively. The percentage of false negative outcome can be regarded as moderately high. This indicates that when a patient is predicted to have failure of the surgery, clinicians need to be very cautious about the predicted results and should make the final decision based on their experience and other clinical examinations. This may avoid the false negatives to a large extent. In addition, as previously discussed, a false-negative outcome might entail incorrectly waiving MMA as the therapy of choice. However, the disadvantages of a false-positive outcome resulting in the incorrect indication for MMA may be more severe when compared to the incorrect waiving of MMA.

In order to increase surgical success rates, a prediction tool is warranted that aids surgeons in identifying responders and non-responders pre-operatively during patient counseling. If a patient is predicted to have a high probability of surgical success, this endorses the consideration for MMA as the therapy of choice. In addition, if a patient is predicted to have a low probability of surgical success, this will aid clinician and patient to be more cautious in choosing MMA as the therapy of choice and possibly search for other therapeutic options. When a patient with a low probability of surgical success is still determined to undergo MMA since he/she has no other therapeutic options left, the prediction might still help to inform the patient on the prognosis of their OSA, thereby shaping their expectations of MMA. The prediction model allows patients to be informed on their individual chances of surgical success rather than average group success rates.

For the presented study population, 67% of the included patients attained surgical success after MMA. These results are lower when compared with a recent review reporting surgical success rates of up to 85% [12]. We believe this is due to the fact that the patients included in this study had more multi-therapy resistant (complex) types of OSA, since these patients were referred to our academic hospital after the failure of one or more earlier therapies. This study included patients with moderate to severe OSA. This is because patients with mild OSA generally experience milder symptoms and therefore a lower burden of disease and a lower risk of untreated hypoxic burden compared to patients with moderate or severe OSA. Therefore, an invasive therapy such as MMA is not considered the therapy of choice in patients with mild OSA, and non-invasive therapies (i.e., CPAP or MAD therapy) resolve symptoms and obtain success of therapy in most cases [9]. The prediction model presented in this study can therefore solely be utilized for patients with moderate to severe OSA.

This study has some limitations. First, the retrospective design of the study entails higher proportions of missing data. The missing data was considered missing at random, and therefore the multiple imputation technique was used for the missing values. Ideally, a prospective study is preferred due to better control of the data. However, since imputation of missing values is considered superior to complete case analysis in the event of missing data, the potential bias in the results caused by the missing values were minimized [43]. Second, in a multivariate logistic regression analysis, an events per variable (EPV) value of 10 is widely advocated to obtain a reliable outcome [44,45]. The present study, however, did not meet the criterion because of the small sample size, which is a limitation. In order to reduce the number of predictors included in the multivariate analysis, we performed univariate analyses to pre-screen the predictors in the study. In addition, we used a less stringent threshold of *p*-value = 0.20 in modeling for the selection of potential predictors to avoid the incorrect exclusion of the important predictors due to the small sample size. In this way, the negative consequence caused by the sample size could be reduced to a large extent. Third, the cephalograms that were assessed in this study were all performed while the patients were awake and with a standard upright position. The data obtained on soft tissue measurements might therefore not be an accurate resemblance of the measurements of soft tissue during sleep in supine position. Nevertheless, it has been performed widely as a routine application prior to OSA surgery, and in the context of low costs and convenience, determining pharyngeal and skeletal anatomy by a cephalogram performed in the standard upright position is of added value. Because we did not have a different population, external validation of the model was not possible in our study, which is a limitation. Therefore, we recommend that the external validation of the model is warranted for future research. Fourth, the postoperative PSG was performed at the minimum of 3 months and at the maximum of 12 months. This difference in the timing of the follow-up PSG might influence the observed success rates of the patients, thus causing a bias in the results. However, several studies have illustrated that the decrease in AHI, and therefore surgical success, after MMA is stable over time [23,46], and it is therefore not likely that the postoperative PSG timing biased the final results in a major way. Last, the missing proportion of the DISE variables was 36%, which is relatively large. The main reason for the absence is that the DISE variables were not routinely collected in the clinical practice, and the variables were more likely to be collected when other alternative treatments for CPAP or MAD were indicated, when surgical options were indicated, or when the AHI was very high and initial therapy did not work. Therefore, we think the DISE variables are likely to be missing not at random, because the factors which may impact the absence of the variables were not adjusted in the imputation model. This may, to some extent, bias our results, which is another limitation.

## 5. Conclusions

The prediction model was developed for the surgical success of MMA as a surgical treatment for patients with moderate to severe OSA. A lower age at surgery, CAI < 5 events per hour, a lower anterior lower face height, and a smaller superior posterior airway space were significant predictors for the surgical success of MMA. The performance of the model terms of discrimination and calibration was acceptable. The clinical added values of the model were adequate for ruling in and ruling out surgical success of treatment. The model presented in this study may aid surgeons in identifying responders for MMA preoperatively. In addition, it improves preoperative patient counseling on the chances of reaching surgical success. However, prior to the implementation of the model in daily clinical practice, external validation is warranted.

## Figures and Tables

**Figure 1 jcm-12-00503-f001:**
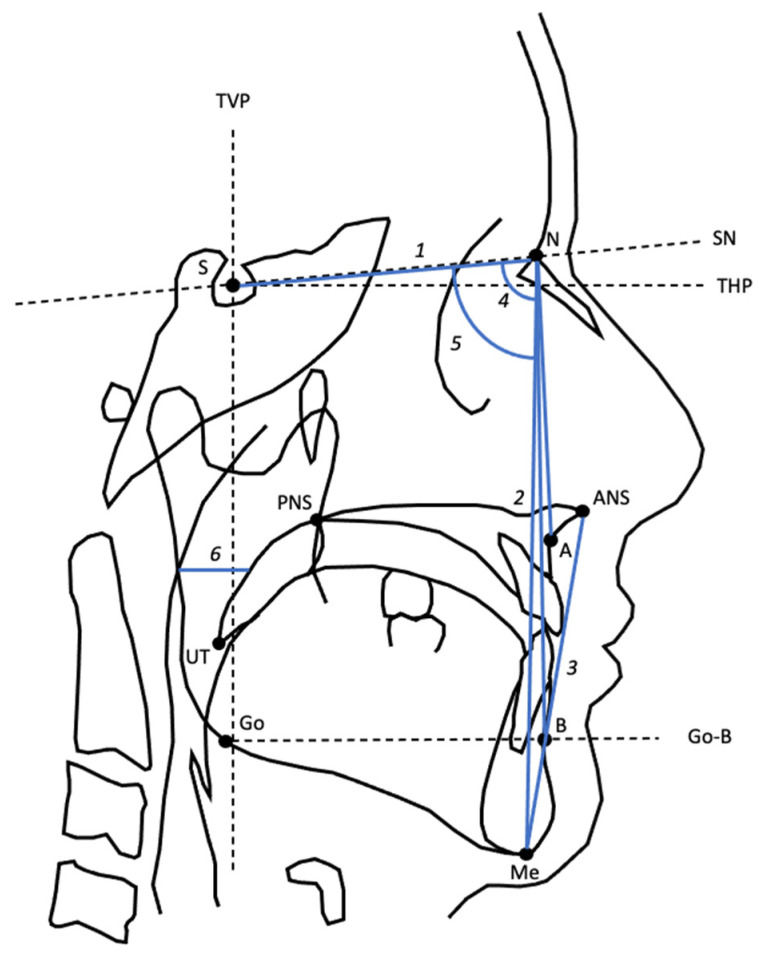
Landmarks, reference lines, and variables used from cephalometry. Landmarks: A, subspinale; ANS, anterior nasal spine; B, supramentale; Go, gonion; Me, menton; N, Nasion; PNS, posterior nasal spine; S, sella; UT, uvula tip; Reference lines: Go-B, gonion-supramentale; SN, sella-nasion; THP, true horizontal plane (through S, set at 7° from SN); TVP, true vertical plane (through S, set at 90° from THP). Variables: 1, S-N; 2, ATFH (anterior total face height, N-Me); 3, ALFH (anterior lower face height, ANS-Me); 4, SNA angle; 5, SNB angle; 6, SPAS (superior posterior airway space).

**Figure 2 jcm-12-00503-f002:**
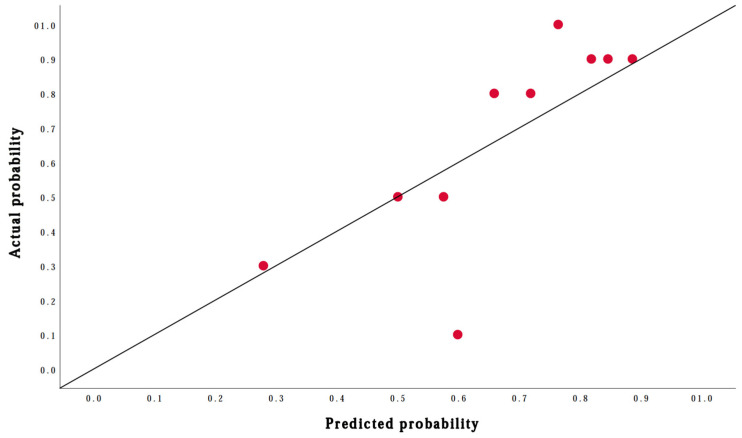
Calibration plot of the prediction model for surgical success. The diagonal line is what would result if the predicted probability of the model was the same as the actual probability of the model so that the prediction is neither underestimated nor overestimated. The red dots represent the deciles of the patients based on their predicted probabilities.

**Figure 3 jcm-12-00503-f003:**
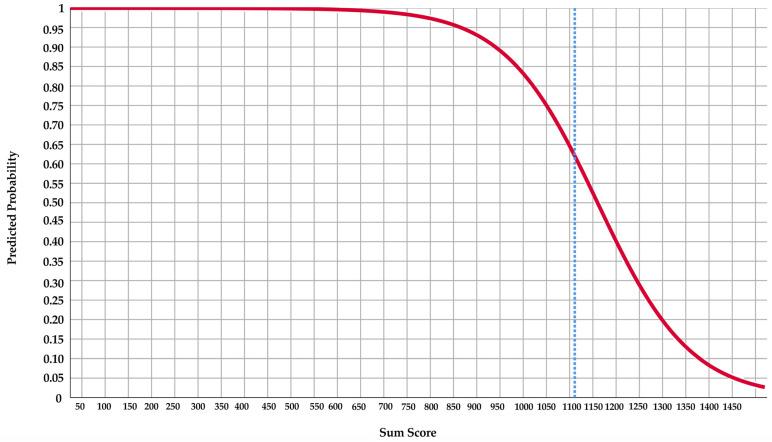
The line chart of the prediction model for surgical success. From the line chart, the exact predicted probability (%) of surgical success of an individual (Axis Y) can be determined based on the sum scores (Axis X) and the curve.

**Table 1 jcm-12-00503-t001:** Cephalometric variables with definitions.

Variable	Definition
S-N	Distance between S and N
ATFH	Distance between N and Me
ALFH	Distance between ANS and Me
SNA	Angle from S to N to A
SNB	Angle from S to N to B
SPAS	Width of the posterior airway at the level of the midpoint of UT and PNS, parallel to line Go-B.

A, subspinale; ALFH, anterior lower face height; ANS, anterior nasal spine; ATFH, anterior total face height; B, supramentale; Go, gonion; Me, menon; N, nasion; PNS, posterior nasal spine; S, sella; S-N, sella-naison line; SPAS, superior posterior airway space; UT, uvula tip.

**Table 2 jcm-12-00503-t002:** Characteristics of the predictors and their distribution over the outcome based on the multiple imputation (*n* = 100).

Potential Predictors (*n* = 100)		Number (%) or Mean (±SD)	Surgical Success	
			Yes (±SD/%) (*n* = 67)	No (±SD/%) (*n* = 33)
Age		50.5 (9.9)	49.3 (9.8)	53.1 (9.7)
Gender	Male	82 (82.0)	54 (80.6)	28 (84.8)
	Female	18 (18.0)	13 (19.4)	5 (15.2)
BMI *		29.7 (27.4–32.2)	29.7 (27.4–32.4)	29.8 (28.2–32.0)
ASA classification score	I	23 (23.0)	17 (25.4)	6 (18.2)
	II	56 (56.0)	38 (56.7)	18 (54.5)
	III	21 (21.0)	12 (17.9)	9 (27.3)
Previous upper airway surgery	Yes	42 (42.0)	27 (40.3)	15 (45.5)
	No	58 (58.0)	40 (59.7)	18 (54.5)
Dentulous (full + partially) **	Yes	82.6 (82.6)	55.6 (83.0)	27 (81.8)
	No	17.4 (17.4)	11.4 (17)	6 (18.2)
Polysomnographic variables				
AHI pre-operative		52.9 (21.4)	54.2 (20.9)	50.3 (22.6)
Positional dependent OSA **	Yes	43.9 (43.9)	29.5 (44)	14.5 (43.9)
	No	56.1 (56.1)	37.5 (66)	18.5 (56.1)
CAI ≥ 5 events/hour **	Yes	16 (16)	7 (10.4)	9 (27.3)
	No	84 (84)	60 (89.6)	24 (72.7)
Cephalometric variables				
Anterior total face height **		123.9 (8.3)	122.8 (7.7)	126.1 (9.2)
Anterior lower face height **		73.0 (7.4)	72.0 (7.2)	75.0 (7.7)
SPAS **		8.3 (2.9)	7.7 (2.7)	9.6 (3.3)
Presence of maxillomandibular deficiency **	Yes	75.4 (75.4)	50.5 (75.4)	24.9 (75.5)
	No	24.6 (24.6)	16.5 (24.6)	8.1 (24.5)
DISE variables				
Concentric collapse velum **	Yes	30.5 (30.5)	17.7 (26.4)	12.9 (39.1)
	No	69.5 (69.5)	49.3 (73.6)	20.1 (60.9)
Complete anteroposterior epiglottis collapse **	Yes	24.2 (24.2)	15.9 (23.7)	8.3 (25.2)
	No	75.8 (75.8)	51.1 (76.3)	24.7 (74.8)

AHI, apnea hypopnea index; ASA, American Society of Anesthesiologists; BMI, body mass index; CAI, central apnea index; DISE, drug-induced sleep endoscopy; OSA, obstructive sleep apnea; SPAS, superior posterior airway space; * values not normally distributed given as median and interquartile range (Q1-Q3)**;** ** including imputed data due to missing values.

**Table 3 jcm-12-00503-t003:** Univariate and multivariate logistic regression analyses for the surgical success (*n* = 100).

Univariate Logistic Regression					Multivariate Logistic Regression			
Predictors	Coding	B (SE)	OR (95%CI)	*p*-Value	B (SE)	Shrunken B	OR (95%CI)	*p*-Value
Intercept					14.258 (5.082)	11.6005		<0.01
Age		−0.041 (0.023)	0.959 (0.917–1.003)	0.070	−0.041 (0.027)	−0.033	0.96 (0.91–1.01)	0.13
Gender	Female Male	Ref. −0.299 (0.575)	0.742 (0.240–2.291)	0.604				
BMI		−0.004 (0.051)	0.996 (0.901–1.101)	0.941				
ASA classification score	I II III	Ref. −0.294 (0.554) −0.754 (0.648)	0.745 (0.251–2.209) 0.471 (0.132–1.676)	0.596 0.245				
Previous upper airway surgery	No Yes	Ref. −0.211 (0.429)	0.810 (0.349–1.879)	0.623				
Dentulous (full + partially)	No Yes	Ref. 0.082 (0.560)	1.085 (0.362–3.252)	0.884				
AHI pre-operative		0.009 (0.010)	1.009 (0.989–1.029)	0.389				
Positional dependent OSA	No Yes	Ref. 0.002 (0.451)	1.002 (0.414–2.428)	0.996				
CAI ≥ 5 events/hour	No Yes	Ref. −1.185 (0.636)	0.306 (0.088–1.065)	0.063	Ref. −1.830 (0.865)	−1.473	0.16 (0.03–0.91)	0.04
Anterior total face height		−0.048 (0.028)	0.953 (0.901–1.008)	0.091				
Anterior lower face height		−0.056 (0.032)	0.945 (0.888–1.006)	0.075	−0.071 (0.036)	−0.057	0.93 (0.87–1.00)	0.05
SPAS		−0.235 (0.083)	0.791 (0.672–0.931)	0.005	−0.280 (0.099)	−0.225	0.76 (0.62–0.92)	0.01
Presence of maxillomandibular deficiency	No Yes	Ref. −0.016 (0.558)	0.984 (0.329–2.945)	0.978				
Concentric collapse velum	No Yes	Ref. −0.587 (0.535)	0.556 (0.194–1.591)	0.273				
Complete anteroposterior epiglottis collapse	No Yes	Ref. −0.050 (0.612)	0.951 (0.285–3.169)	0.935				

AHI, apnea hypopnea index; ASA, American Society of Anesthesiologists; BMI, body mass index; CAI, central apnea index; DISE, drug-induced sleep endoscopy; OR, odds ratio; OSA, obstructive sleep apnea; SE, standard error; SPAS, superior posterior airway space.

**Table 4 jcm-12-00503-t004:** Clinical (added) values of the model (*n* = 100).

Outcome	Prevalence (95% CI)	Sensitivity (95% CI)	Specificity (95% CI)	PPV (95% CI)	NPV (95% CI)	Added Value for Ruling in the Outcome (95% CI)	Added Value for Ruling Out the Outcome (95% CI)
Surgical success	0.67 (0.57–0.76)	0.79 (0.68–0.88)	0.79 (0.62–0.90)	0.88 (0.78–0.95)	0.65 (0.49–0.79)	0.21 (0.09–0.34)	0.32 (0.15–0.49)

CI, confidence interval; NPV, negative predictive value; PPV, positive predictive value.

**Table 5 jcm-12-00503-t005:** Score chart for the prediction of surgical success.

Predictors		Score
Anterior lower face height		6
SPAS		23
Age		3
CAI ≥ 5 events/hour	No	0
	Yes	147
Sum score		

CAI, central apnea index; SPAS, superior posterior airway space.

## Data Availability

Not applicable.

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
