# Peer review of "Development and Internal Validation of a Prediction Model for Surgical Success of Maxillomandibular Advancement for the Treatment of Moderate to Severe Obstructive Sleep Apnea"

_jcm, 2023, doi:10.3390/jcm12020503_

Round 1

Reviewer 1 Report

I congratulate you on your excellent work.

Please answer some queries that I have. The confidence intervals for ORs for "age" and "anterior lower face height" are spanning the null value of 1. Would it be appropriate to include them in the mathematical model. Also, the robustness of the model is questionable considering a significance level of 0.2 has been chosen.

There seems to be some typing error in the CAI events in table 2 (numbers do not add up)

Reviewer 2 Report

The aim of this study was to develop, internally validate, and calibrate a prediction model for presence of surgical success for maxillomandibular advancement (MMA) in patients with OSA.  The authors show that lower age at surgery, CAI <5 events/hour, lower ALFH, and smaller SPAS were significant predictors for surgical success of MMA. Overall, the study is well conducted and the results are interesting. I've only minor comments that are related to additional limitations that need to be addressed.

1. The authors state in the Methods section: "The present study was carried out based on ... Transparent Reporting of a multivariable prediction model for Individual Prognosis or Diagnosis (TRIPOD) statement (17)." Usually, the development of the model is made on a population, while the validation of the models is made on another population. For instance in TRIPOD checklist it is stated: "Development vs. validation: For validation, identify any differences from the development data in setting, eligibility criteria, outcome, and predictors." Given the restricted size of the population, I understand that two sets of patients cannot be made, but the authors should clearly discuss this limitation in the Discussion section, and should state that an independent set of patients is warranted for validation.

2. Methods: the authors state "The surgical success of MMA is considered ‘present’ if a patient’s AHI reduced by ≥50% compared to the preoperative AHI, combined with a postoperative AHI <20 events/hour. The criteria for surgical success was based on Sher’s criteria for determining surgical success for OSA (15)." The authors should simplify the sentence as: ... < 20 events/hour, as proposed by Sher and colleagues (15). Please correct ref 15: names do not appear.

3. Methods: the exclusion criteria are non-inclusion criteria (since the participants have not been previously included): non-inclusion instead of exclusion.

4. Overall, the independent factors associated with the favourable outcome may seem expected. In the Supplemental table it would be interesting to give the mean central apnea index. One may wonder whether some patients had MMA while suffering from predominant central apnea syndrome.
